# Approximating multivariate posterior distribution functions from Monte Carlo samples for sequential Bayesian inference

Bram Thijssen[1,2], Lodewyk F. A. Wessels[1,2,3]*

1 Division of Molecular Carcinogenesis, Netherlands Cancer Institute, Amsterdam, The Netherlands,
2 Oncode Institute, Amsterdam, The Netherlands, 3 Faculty of Electrical Engineering, Mathematics and Computer Science, Delft University of Technology, Delft, The Netherlands

* l.wessels@nki.nl

**Data Availability Statement:** All relevant data are within the paper and its Supporting Information files.

**Funding:** This work was performed within the Cancer Genomics Netherlands Program supported

## Abstract

An important feature of Bayesian statistics is the opportunity to do sequential inference: the posterior distribution obtained after seeing a dataset can be used as prior for a second inference. However, when Monte Carlo sampling methods are used for inference, we only have a set of samples from the posterior distribution. To do sequential inference, we then either have to evaluate the second posterior at only these locations and reweight the samples accordingly, or we can estimate a functional description of the posterior probability distribution from the samples and use that as prior for the second inference. Here, we investigated to what extent we can obtain an accurate joint posterior from two datasets if the inference is done sequentially rather than jointly, under the condition that each inference step is done using Monte Carlo sampling. To test this, we evaluated the accuracy of kernel density estimates, Gaussian mixtures, mixtures of factor analyzers, vine copulas and Gaussian processes in approximating posterior distributions, and then tested whether these approximations can be used in sequential inference. In low dimensionality, Gaussian processes are more accurate, whereas in higher dimensionality Gaussian mixtures, mixtures of factor analyzers or vine copulas perform better. In our test cases of sequential inference, using posterior approximations gives more accurate results than direct sample reweighting, but joint inference is still preferable over sequential inference whenever possible. Since the performance is case-specific, we provide an R package *mvdens* with a unified interface for the density approximation methods.

## Introduction

In Bayesian statistics, unknown variables are given a probability distribution that specifies our knowledge about the variables. This distribution can then be updated based on available data using Bayes' theorem. An important advantage of this approach is that inference can be done sequentially; that is, when we have obtained a posterior distribution after seeing a first dataset, we can use this posterior as prior for inference with a next dataset.

by the Gravitation program of the Netherlands Organization for Scientific Research (NWO). The funders had no role in study design, data collection and analysis, decision to publish, or preparation of the manuscript.

**Competing interests:** The authors have declared that no competing interests exist.

For models where the posterior distribution is not analytically tractable, Bayesian inference is often achieved with some variant of Monte Carlo sampling. This allows us to obtain samples from posterior distributions. When we want to use the Monte Carlo sampling results for sequential inference, we only have this set of samples to use as prior. We can use these samples directly for sequential inference, by reweighting them accordingly, but the sequential posterior will then only be evaluated at those sample points, which may not be accurate. Alternatively, we can estimate a functional representation of the first posterior, and use this functional representation as prior for the second inference, and proceed with any Monte Carlo sampling scheme as usual.

There are various situations where sequential inference may be useful. For example, it can be conceptually appealing to summarize the posterior of one dataset and continue inference with a second dataset without having to refer back to the first. As an example of this, in astronomy, Wang et al. [1] have estimated posterior distributions for orbital eccentricities which can then subsequently be used as prior in further research. Alternatively, a modeler may have fitted a model to a dataset, and when additional data arrives he or she may wish to update the posterior with the new data. Often the inference is a time-consuming process [2–5], and it is not always feasible to do a new joint inference each time new data arrives. Efficiency might also be gained in special instances, for example when parameters can be dropped for parts of the data.

We therefore wished to investigate whether sequential inference is a feasible approach, even when using Monte Carlo sampling for the separate inference steps. Specifically, we wished to test whether we can obtain an accurate joint posterior $P(\mathbf{x}|y_1, y_2)$ from two datasets, by first sampling the posterior of one dataset and then performing sequential inference with the second dataset using an approximation of the first posterior as prior:

$$P(\mathbf{x}|y_1, y_2) \approx \frac{P(y_2|\mathbf{x})\hat{P}(\mathbf{x}|y_1)}{P(y_2)}.$$

Here $\mathbf{x}$ is a vector of the variables of interest, $y_1$ and $y_2$ are two datasets, and $\hat{P}$ is an approximation of the posterior of the first dataset obtained from Monte Carlo samples (see Methods section). Throughout this article we assume that datasets are independent given the model. It is important to note that doing statistical inference with multiple datasets may require additional parameters or a hierarchical structure to account for differences between datasets. We will explicitly mention when we use dataset-specific parameters and when we will assume them to be the same between datasets.

Estimating functional forms of posterior distributions from Monte Carlo samples is an established part of Bayesian analysis [6], and could be done with a large variety of methods. Broadly, this might be done in two ways. One option is to treat the posterior distribution approximation task as a general density estimation problem, where we estimate the density function only from the location of the samples. Several popular density estimation methods include kernel density (KD) estimation [7], Gaussian mixtures (GM) [8], mixtures of factor analyzers (MFA) [9], and copulas or vine copulas (VC) [10]. An alternative option is to treat the posterior distribution approximation task as a regression problem, since alongside the sample positions, we usually also have the relative value of the posterior probability at the sample locations. This has the advantage of using additional information of the posterior distribution, but presents its own challenges as well. In particular, the regression function must integrate to one for it to be a proper density function. It can be challenging to meet this constraint while fitting a function through many sample points. One regression method with sufficient flexibility to achieve this is Gaussian process (GP) regression [11].

To test our question of whether sequential inference can be done by estimating a functional approximation of the first posterior, we will consider each of the aforementioned methods (density estimation with KDs, GMs and VCs, and regression with GPs). We first test their performance in approximating a known density, then test their accuracy in approximating a posterior distribution from Monte Carlo samples, and subsequently test their performance in sequential inference. Finally, we test whether sequential inference of two datasets is computationally faster than inference with the two datasets jointly.

Besides in sequential inference, posterior distribution approximations are also used in several other areas of Bayesian computation. First, in Monte Carlo sampling itself, a proposal distribution is used, and sampling is most efficient when the proposal distribution resembles the true target probability density. There have been many efforts to create efficient proposal distributions, including using some of the density approximation methods that we consider here, for example with vine copulas [12] and Gaussian processes [13]. Second, posterior distribution approximations have been used in schemes for parallelizing MCMC inference [14]. In this case the inference is split into parts, and the resulting subposteriors are combined using a posterior distribution approximation to recover the full posterior. Third, in the area of Bayesian filtering [15], a posterior distribution is updated when new data arrives over time, which also relies on posterior distribution approximations. In the present study, we explicitly test the accuracy in approximating posterior distributions, and, apart from the use of such approximations in sequential inference, the results presented here may be relevant for these other areas as well.

## Methods

To use the posterior obtained from Monte Carlo sampling in sequential inference, we need to approximate the distribution

$$P(\mathbf{x}|y) = \frac{P(y|\mathbf{x})P(\mathbf{x})}{P(y)} \approx \hat{P}(\mathbf{x}),$$

where $\mathbf{x}$ is the $D$-dimensional variable of interest and $y$ represents the inference data. In the notation of the approximation $\hat{P}(\mathbf{x})$ we have dropped the conditioning on $y$ for brevity.

The approximation $\hat{P}(\mathbf{x})$ needs to be constructed from samples $\mathbf{x}_i$ that have been drawn from the posterior $P(\mathbf{x}|y)$. The approximations can be achieved using density estimation or through regression, see Fig 1. In all subsequent equations, $N$ is the number of Monte Carlo samples and $\mathbf{x}_i$ is the $D$-dimensional value of the $i$th sample. While $i$ indexes the samples, $j$ indexes the dimensions, so note that $x_j$ (non-bold, and indexed by $j$) refers to the $j$th element of the $D$-dimensional value of $\mathbf{x}$. For the regression methods, we assume that the relative, unnormalized probability is available, and it is represented by $p_i$ for sample $i$, (that is, $p_i = P(y|\mathbf{x}_i)P(\mathbf{x}_i)$).

### Density estimation

The density estimation methods use the sample positions, $\mathbf{x}_i$, to reconstruct an approximation to the probability density function. Below we briefly introduce three density estimation methods: kernel density estimation, Gaussian mixtures and vine copulas, with several variations. Fig 2 shows an example of each method in a bivariate case.

**Kernel density.**   The kernel density approximation is given by

$$\hat{P}_{KD}(\mathbf{x}) = \frac{1}{N}\sum_{i=1}^{N} K(\mathbf{x} - \mathbf{x_i}),$$

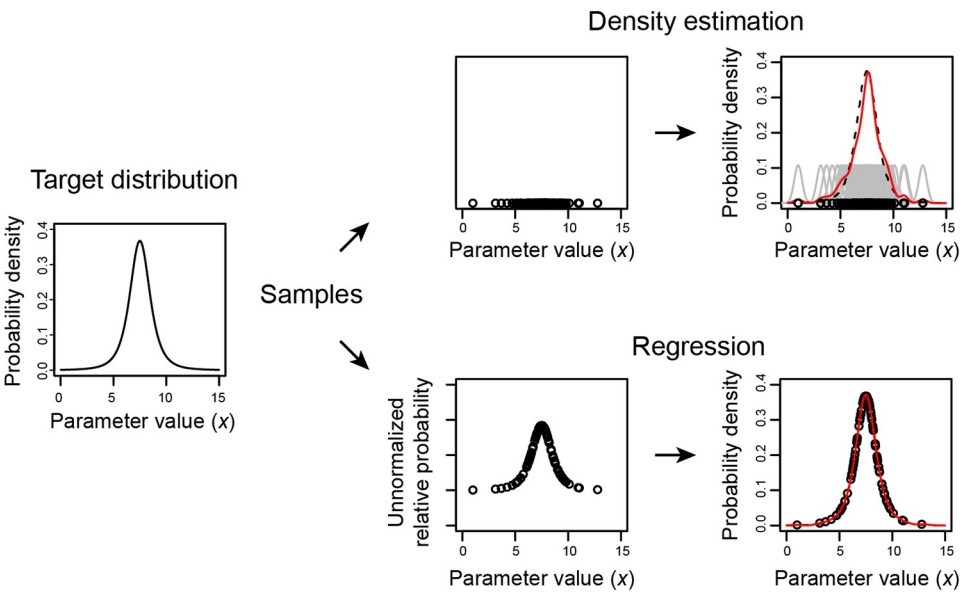

**Fig 1. Reconstructing a probability density function by density estimation or regression.** Using Monte Carlo sampling we have samples drawn from the target probability distribution. With density estimation, the locations of the samples is used to reconstructed the probability distribution. With regression, both the sample location and the unnormalized, relative probability is used to reconstruct the probability distribution. The example function is a *t*-distribution with $v = 4$ centered at 7.5.

where $K(\mathbf{x} - \mathbf{x_i})$ is a kernel function. We take the kernel function to be a multivariate normal distribution $\mathcal{N}(0, \Sigma)$. When $D \leq 6$ we estimate a full covariance matrix using multivariate plug-in bandwidth selection [16] as implemented in the *R* package *ks* [17]. When $D > 6$ we estimate a diagonal covariance matrix with the diagonal entries using Scott's reference rule [7]; that is, the empirical standard deviation multiplied by $N^{-1/(D+4)}$.

**Gaussian mixture.** The Gaussian mixture approximation is given by

$$\hat{P}_{GM}(\mathbf{x}) = \sum_{g=1}^{G} c_g \mathcal{N}(\mathbf{x}|\mu = \boldsymbol{\mu}_g, \Sigma = \Sigma_g),$$

where $c_g$, $\boldsymbol{\mu}_g$ and $\Sigma_g$ are the proportion, mean and covariance of the *g*th component, *G* is the number of mixture components, and $\Sigma c_g = 1$. We use a full covariance matrix, and the parameters $c$, $\boldsymbol{\mu}$ and $\Sigma$ are estimated using expectation-maximization. The number of components is selected by minimizing the Akaike information criterion (AIC).

**Truncated gaussian mixture.** When the prior probability distribution $P(\mathbf{x})$ is bounded, we can use truncated Gaussians with known bounds in the mixture:

$$\hat{P}_{TGM}(\mathbf{x}) = \sum_{g=1}^{G} c_g \mathcal{N}_T(\mathbf{x}|\mu = \boldsymbol{\mu}_g, \Sigma = \Sigma_g, a = \mathbf{a}, b = \mathbf{b}),$$

where $\mathbf{a}$ and $\mathbf{b}$ are the known lower and upper bounds respectively. The parameters are estimated using expectation-maximization for truncated Gaussian mixtures [18], using the truncated Gaussian moment calculations provided by [19]. The number of components is selected by minimizing the AIC.

**Mixture of factor analyzers.** In factor analysis, a latent variable $\mathbf{z}$ is introduced to represent the target distribution in a lower-dimensional space. The elements of the *m*-dimensional

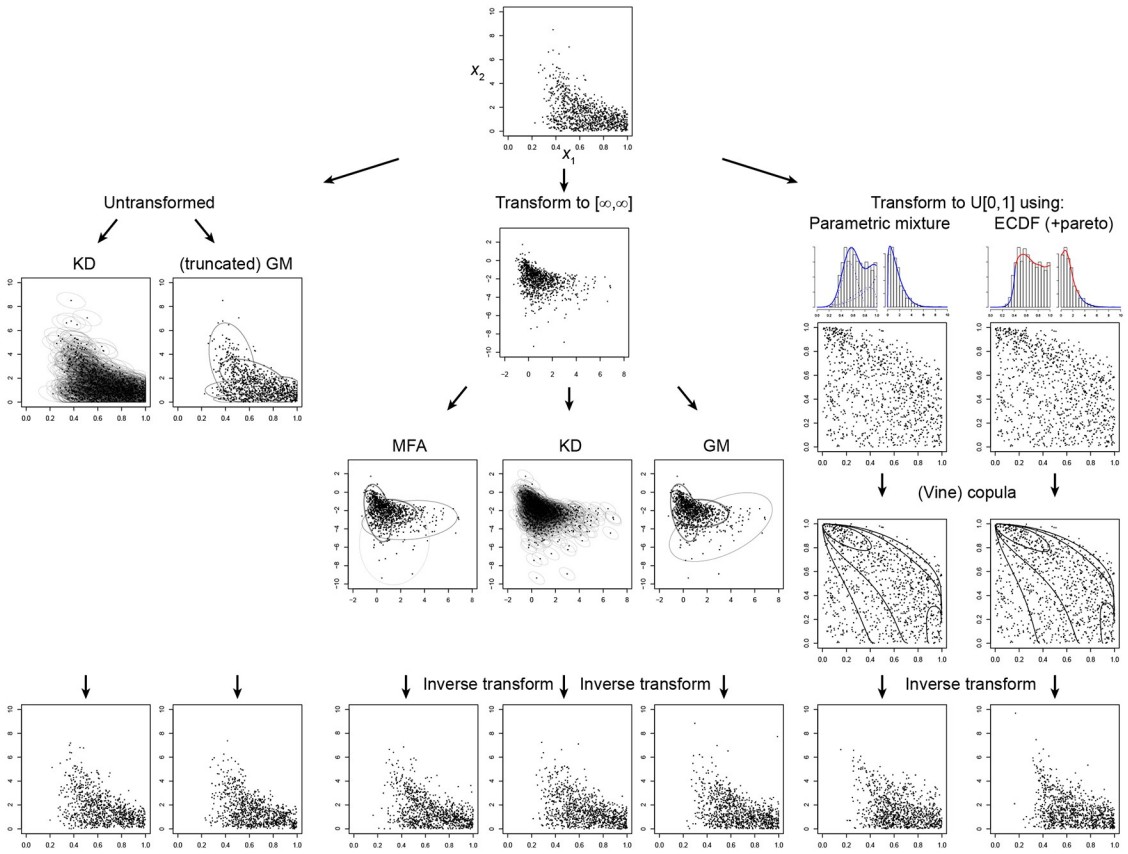

**Fig 2. Density estimation methods applied to a bivariate example.** At the top, we start with samples obtained through Monte Carlo sampling from the posterior of two variables ($x_1$ and $x_2$). The two variables are $\beta_{kill}$ and $\gamma$ from the (bounded) Lotka-Volterra example discussed later. On the left, a kernel density estimate or a Gaussian mixture is fitted to the samples. In the middle, the variables are first transformed to an unbounded domain (in this case through a scaled logit transform) before a KD or GM is fitted. On the right, the variables are transformed to have uniform marginal distributions between 0 and 1, using either a parametric mixture or an empirical cumulative distribution with Pareto tails. Subsequently, a copula function is fitted to the transformed variables. Finally, on the bottom row, new samples are drawn from each of the approximations. Where necessary, the new samples are transformed with the inverse of the original transformation. In each case the distribution of the new samples is similar to the original sample distribution, but slight differences between the approximations can be observed as well.

variable **z** are called the factors. This can be extended to mixtures of factor analyzers in multiple ways, one of which consists of introducing multiple factor analyzers to describe different parts of the target distribution [9]. The approximation of our posterior distribution is then given by

$$\hat{P}_{MFA}(\mathbf{x}) = \sum_{g=1}^{G} c_g \int \mathcal{N}(\mathbf{x}|\mu = \boldsymbol{\mu}_g + B_g\mathbf{z}, \Sigma = \Psi_g)P(\mathbf{z})d\mathbf{z}$$

where **z** is the $m$-dimensional latent variable, $B_g$ are the $D \times m$ loading matrices, $\Psi_g$ are diagonal covariance matrices with elements $\sigma_1, \ldots \sigma_D$ and $c_g$ are the mixture weights. The factors **z** are assumed to be distributed by a unit normal distribution with zero mean. For simplicity, we only consider the case where each mixture element has a separate mean, loading matrix and covariance matrix. The mixtures of factor analyzers are fitted using the Alternating Expectation Conditional Maximization algorithm [9], as implemented in the *R* package *EMMIXmfa* [20]. The number of components and number of factors are selected by minimizing the AIC.

**Vine copula.** With copulas, the multivariate distribution is decomposed into marginal distributions and a description of the dependency structure. The copula density approximation is then given by

$$\hat{P}_{COP}(\mathbf{x}) = c(F_1(x_1), \ldots, F_D(x_D)) \prod_{j=1}^{D} f_j(x_j),$$

where $c$ is a copula function, $f_j$ is the marginal probability density function for dimension $j$, and $F_j$ the corresponding marginal cumulative density function. Various different families of copula function exist; by using the *R* package *VineCopula* [21], we evaluate various commonly used families and their rotations and select the optimal function by minimizing the AIC.

For $D > 2$; a multi-dimensional copula function could be used, but we instead model the approximation using regular vine copulas [22], given by the equation

$$\hat{P}_{VC}(\mathbf{x}) = \prod_{l=1}^{D-1} \prod_{k=1}^{D-l} c_{k,(k+l)|(k+1),\ldots,(k+l-1)} \prod_{j=1}^{D} f_j(x_j),$$

where the first two products are the pair-copulas and the third product contains the marginal densities as before. The bivariate pair-copula functions are selected as before by minimizing the AIC, and the vine structure is selected using a maximum spanning tree with Kendall's tau edge weights [23].

For the marginal distribution and density functions, common choices include empirical distribution functions and parametric distributions. We consider these two options, as well as using Pareto tails and parametric mixtures:

- *Empirical distribution marginal*: An empirical marginal distribution function is given by

$$F_j(x_j) = \frac{1}{N} \sum_{i=1}^{N} \mathbf{1}_{x_{i,j} \leq x_j},$$

where $\mathbf{1}_{x_{i,j} \leq x_j}$ is the indicator function. A corresponding density function is constructed using a 1-dimensional kernel density estimate

$$f_j(x_j) = \frac{1}{N} \sum_{i=1}^{N} \mathcal{N}(x_{i,j}, \sigma_j),$$

where $\sigma_j$ is estimated using plug-in bandwidth selection.

For the quantile function (the inverse of the cumulative distribution function) we use a linear interpolation of the empirical distribution function. When the prior has bounded support, samples are mirrored across the boundary to improve the estimate near the boundaries.

- *Pareto tails*: Since an empirical distribution can be inaccurate in the tails, we also consider augmenting the empirical density with Pareto tails. The distribution is then split in three parts, a body described by the empirical distribution function and kernel density estimate as before, and two tails described by a generalized Pareto distribution (GPD). An important choice is where to put the threshold beyond which data are used to fit the tail distribution [24]. We use the simple rule of thumb of using 10% of the samples to estimate a tail [25]. Since we have a tail on each side, we use the middle 80% of the samples for the body, and the

upper and lower 10% of the samples to estimate the Pareto tail on each side:

$$F_j(x_j) = \begin{cases} q - qF_{j,\xi_{j,1}}\left(\dfrac{t_{j,1} - x_j}{\sigma_{j,1}}\right) & \text{if } x_j \leq t_{j,1} \\[2mm] (1-q) + qF_{j,\xi_{j,2}}\left(\dfrac{x_j - t_{j,2}}{\sigma_{j,2}}\right) & \text{if } x_j \geq t_{j,2} \\[2mm] F_{j,\text{ECDF}}(x_j) & \text{otherwise,} \end{cases}$$

where

$$F_\xi(z) = 1 - (1 + \xi - z)^{-1/\xi}$$

is the GPD function, $q$ is the quantile used for the threshold ($q = 0.1$ for the 10% rule), and $t_{j,1}$ and $t_{j,2}$ are the lower and upper $q$th quantile of $x_j$ respectively. $F_{j,\text{ECDF}}(x_j)$ is the empirical distribution function as before. To ensure continuity in the density function between the Pareto tail and the ECDF body, we set $\sigma_j = q/f_{j,KD}(t_j)$. The shape parameter $\xi_j$ is estimated by maximum likelihood, separately for each tail. The density function of the tails is given by the GPD density, scaled by $q$:

$$f_\xi(z) = \frac{q}{\sigma_j}(\xi z + 1)^{-(\xi+1)/\xi}.$$

In the case of bounded support, we do not use a Pareto tail unless the empirical density at the boundary is less than a threshold $\epsilon$ (which we set to $1/N$). While a GPD can handle a bounded support (by taking $\xi < 0$), we find this often leads to a poorer approximation than an empirical estimate with mirroring across the boundary.

- *Parametric mixtures*: The marginal densities can also be approximated with mixtures of parametric distributions. For unbounded variables we use a mixture of normals:

$$f_j(x_j) = \sum_{g=1}^{G} c_g \mathcal{N}(x_j | \mu = \mu_g, \sigma^2 = \sigma_g^2).$$

When there are known bounds, we use gamma distributions (when there is only a lower or upper bound) or beta distributions (when there is both a lower and upper bound) instead of normal distributions; these distributions are scaled, shifted and/or reflected to match the bounds. The parameters are estimated using expectation-maximization, and we select the number of components by minimizing the AIC.

## Regression

When the relative probability density at the sample positions is available, the density function can be estimated by regression. Typically, only the relative, unnormalized probability density will be available. If that is the case, it will be necessary to normalize the regression function to ensure that it integrates to one over the prior domain.

When an estimate of the marginal likelihood $P(y)$ is available in addition to the samples, then the probability values can be normalized before entering the regression. If the approximation is accurate, this would ensure that the regression function is properly normalized as well, but we don't further explore this option of normalization with a known marginal likelihood here.

**Gaussian process.** As regression method we employ Gaussian process regression, since it provides flexibility for approximating arbitrary density functions, and it handles multivariate regressors naturally. In order to handle unnormalized input densities, we multiply the Gaussian process predictive distribution with a scaling parameter. By calculating the integral of the predictive distribution (see below), we can constrain the distribution to integrate to one by setting the scaling parameter to the reciprocal of the integral.

The behavior of Gaussian processes is characterized by their mean and covariance functions. We set the mean function to be zero everywhere, as we expect the probability to go to zero in regions where we do not have any samples. The predictive mean of the Gaussian process function based on the input samples $X$ is then given by:

$$\hat{P}_{GP}(\mathbf{x}) = \frac{1}{Z} K(\mathbf{x}, X) K(X, X)^{-1} \mathbf{p},$$

where $Z$ is the normalizing constant (see below) and $K(X_1, X_2)$ is the matrix obtained by applying the covariance function $k(\mathbf{x_1}, \mathbf{x_2})$ to all pairs of $X_1$ and $X_2$ (see e.g. [11] for more details on Gaussian processes).

As covariance function we consider two commonly used kernels, the squared exponential

$$k_{SE}(\mathbf{x}, \mathbf{x}^*) = \exp\left(\frac{r^2}{2l^2}\right)$$

and the Matérn kernel with $v = {}^3/_2$

$$k_{Mat32}(\mathbf{x}, \mathbf{x}^*) = (1 + \frac{\sqrt{3}r}{l}) \exp\left(-\frac{(\sqrt{3}r)}{l}\right),$$

where $r$ is the Euclidean norm $|\mathbf{x} - \mathbf{x}^*|$ and $l$ a length scale parameter. The parameter $l$ is optimized by minimizing the root mean square error of $\hat{P}_{GP}(\mathbf{x})$ in a 5-fold cross-validation.

In order to normalize the Gaussian process predictive distribution such that it integrates to 1, it is necessary to calculate the integral:

$$Z = \int_{-\infty}^{\infty} K(\mathbf{x}^*, X) K(X, X)^{-1} \mathbf{p} d\mathbf{x}^*.$$

Solving $K(X, X)^{-1} \mathbf{p} = \boldsymbol{\alpha}$, we have

$$\begin{aligned} Z &= \int_{-\infty}^{\infty} K(\mathbf{x}^*, X) \boldsymbol{\alpha} d\mathbf{x}^* \\ &= \int_{-\infty}^{\infty} \sum_{i=1}^{N} k(\mathbf{x}^*, \mathbf{x_i}) \alpha_i d\mathbf{x}^* \\ &= \sum_{i=1}^{N} \alpha_i \int_{-\infty}^{\infty} k(\mathbf{x}^*, \mathbf{x_i}) d\mathbf{x}^* \end{aligned}$$

In the case of the squared exponential kernel $k(\mathbf{x}^*, \mathbf{x}) = \exp\left(-\frac{|\mathbf{x}^* - \mathbf{x}|}{2l^2}\right)$, we have

$$\int_{-\infty}^{\infty} k(\mathbf{x}^*, \mathbf{x}) d\mathbf{x}^* = (\sqrt{2\pi l^2})^D$$

and

$$Z = (\sqrt{2\pi l^2})^D \sum_{i=1}^{N} \alpha_i.$$

For any isotropic kernel $k(\mathbf{x}^*, \mathbf{x}) = h(|\mathbf{x}^* - \mathbf{x}|)$ we can transform to polar coordinates to get

$$\int_{-\infty}^{\infty} h(|\mathbf{x}^* - \mathbf{x}|)d\mathbf{x}^* = \omega_{D-1} \int_{0}^{\infty} h(r)r^{D-1}dr,$$

where $r = |\mathbf{x}^* - \mathbf{x}|$ and $\omega_{D-1}$ is the surface area of a $(D - 1)$-sphere with unit radius, which can be calculated as

$$\omega_{D-1} = \frac{2\pi^{D/2}}{\Gamma\left(\frac{D}{2}\right)}.$$

For the Matérn kernel with $v = \frac{3}{2}$ this gives

$$\int_{-\infty}^{\infty} h(|\mathbf{x}^* - \mathbf{x}|)d\mathbf{x}^* = \frac{2\pi^{D/2}}{\Gamma\left(\frac{D}{2}\right)} \left(\frac{l}{\sqrt{3}}\right)^D (1 + D)\Gamma(D).$$

A downside of using Gaussian processes for probability densities is that they do not naturally allow for a constraint that the function is non-negative everywhere. As a result, negative probability densities can occur. This could be circumvented by transforming the densities (for example by log transform (as done in [26]) or logistic transform (as done in [13])), but then the predictive function can no longer be normalized to integrate to one in the untransformed space. We found that, in our test cases, constraining $Z$ to be positive during the optimization of $l$ prevented large negative densities, and any remaining negative densities were typically very small and were pragmatically set to zero.

## Importance reweighting

As reference for the approximation methods, instead of constructing an approximate distribution function, we can also use the Monte Carlo samples from the initial inference directly and reweight them given the likelihood of the second dataset. That is, the samples are given weights

$$w_i = P(y_2|\mathbf{x}_i)/\sum_{i=1}^{N} P(y_2|\mathbf{x}_i),$$

where $y_2$ indicates the data in the second inference and $\mathbf{x}_i$ are the sample positions from the first inference as before. This can be viewed as importance sampling from the joint posterior distribution with the posterior of the first dataset as proposal distribution, with the fixed set of samples.

## Transformations for bounded variables

Some of the approximation methods can explicitly handle a bounded support. In the other cases, we can use rejection sampling to discard samples outside the prior support. Alternatively, the variables can be transformed to an unbounded domain before applying the posterior approximation methods. We consider a log transform (when there is only a lower or upper bound) or a logit transform (when there is both a lower and upper bound), and scale, shift or

reflect the variables as necessary. The probability density function is corrected for the transformation by multiplying with the derivative of the transform.

## Marginal likelihood estimation from posterior approximation

When the approximation of the posterior distribution function can be normalized such that it integrates to one (as is the case for all methods used here), we can use the approximation to obtain an estimate of the marginal likelihood. Since $\hat{P}(\mathbf{x}) \approx P(\mathbf{x}|y)$, and

$$P(\mathbf{x}|y) = \frac{P(y|\mathbf{x})P(\mathbf{x})}{P(y)},$$

we can use a linear regression of the approximation probability density against the unnormalized posterior probability at each sample position and obtain an estimate $\hat{P}(y)$ of the marginal likelihood from the slope of the regression. Depending on the setting, it may be beneficial to log transform the probabilities:

$$\log \hat{P}(\mathbf{x}) = \log\left(P(y|\mathbf{x})P(\mathbf{x})\right) - \log \hat{P}(y)$$

and get an estimate of the log marginal likelihood from the intercept of the regression.

## Monte Carlo sampling

Unless stated otherwise, Monte Carlo sampling was done using parallel tempered Markov chain Monte Carlo (PT-MCMC) [27] with automated parameter blocking [28]. When using MCMC, the samples are subsampled to produce a chain with no observable autocorrelation. The first half of the simulation is always removed as burn-in, and adaptation is only done during the burn-in period. In the section on marginal likelihood estimation, we also used sequential Monte Carlo (SMC) with MCMC proposal distributions [29], and nested sampling [30]. Marginal likelihood estimates were obtained by thermodynamic integration (when using PT-MCMC), by the resampling weights (when using SMC) and by sampling the mass ratios (when using nested sampling). The sampling and marginal likelihood estimation were done using the Bayesian inference software package BCM [31].

## Implementation—*mvdens*

Implementations of the density approximation methods are available as an R package *mvdens* at http://ccb.nki.nl/software/mvdens.

## Results

### Approximating known target densities

To test whether the density approximation methods can adequately describe multivariate density functions, we first attempted to reconstruct several known target distributions, representing different features which might arise in posterior distributions, namely multimodality, ridges and heavy tails.

**Gaussian mixture.** As first test, we used a mixture of two multivariate Gaussians,

$$P(\mathbf{x}) = \frac{2}{3}\mathcal{N}(\boldsymbol{\mu}_1, \Sigma_1) + \frac{1}{3}\mathcal{N}(\boldsymbol{\mu}_2, \Sigma_2),$$

with random covariance matrices and $\boldsymbol{\mu}_1 = \boldsymbol{\mu}_2 = 0$ for the first test case. Fig 3A (left panel) shows 500 random samples drawn from this distribution for $D = 2$. We then compared how

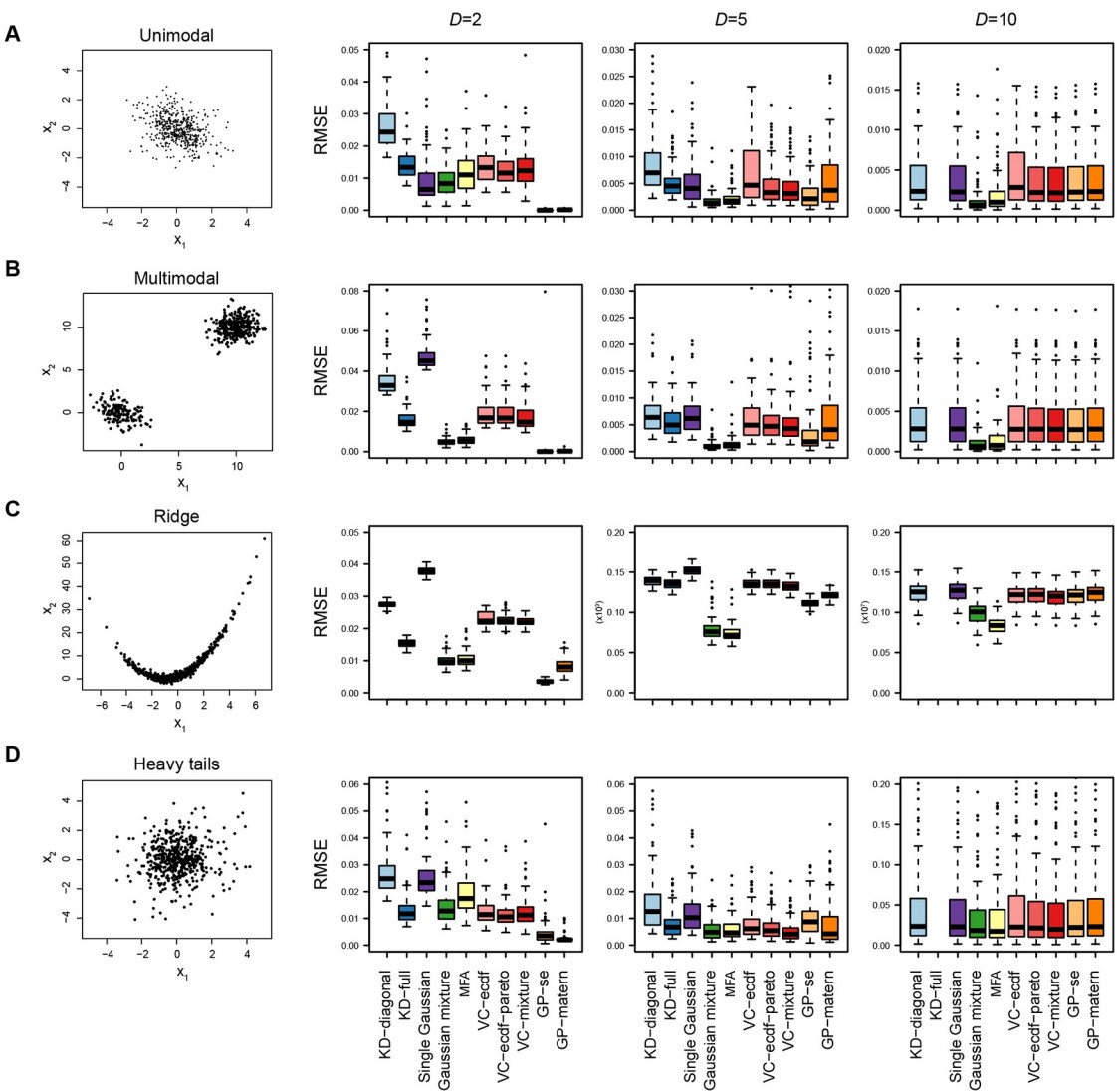

**Fig 3. Comparison of the approximation methods for reconstructing known target distributions with increasing dimensionality.** The density approximations were trained on 500 samples, and the accuracy was evaluated by the root mean square error (RMSE) calculated over 500 new samples. This procedure was repeated 100 times and the boxplots show the resulting RMSEs. Note that the scale of the RMSE is different for different dimensionalities and test cases, as the mean density at the location of the samples is also different.

well the approximation methods can reconstruct this density from 500 samples, at increasing dimensionality (Fig 3A, right panels).

In the low-dimensional setting, Gaussian processes give the best approximation. Since the Gaussian processes can use the relative probability density at the sample positions, they have more information to create a good approximation, which allows a very good reconstruction already with few samples. In the higher-dimensional setting however, the Gaussian processes do not perform as well, likely due to having only a single length scale parameter *l*. Fitting such a regression through high dimensional multivariate sample points leads to an overdispersed distribution, which is limiting the performance.

At *D* = 10, the Gaussian mixture approximation achieves higher accuracy than all other approaches, including Gaussian process regression. Among the density estimation methods, it

is to be expected that the Gaussian mixture approximation is most accurate, since it has the same functional form as the target density.

**Multimodality.** To test the performance of the approximation methods in a multimodal setting, we separated the two Gaussians in space by setting $\mu_2 = \mu_1 + 10$ in every dimension (Fig 3B). As before, Gaussian processes work well in low dimensions, while Gaussian mixtures are better at higher dimensions. In this multimodal case, vine copulas do significantly worse even at $D = 2$. This is likely due to the fact that the available copula functions are designed to describe the shape of a single mode, and are not necessarily suited for describing multimodal distributions. As in the unimodal case, using Pareto tails or parametric mixtures does tend to give slightly better performance than using only an ECDF marginal. For the GP kernels, the squared exponential kernel has better performance than the heavier-tailed Matérn kernel in this case, which is to be expected given the exponential nature of the target distribution.

**Ridges.** Another difficulty which can occur in posterior distributions is the presence of ridges. To test how well the approximation methods can deal with this, we tested a ridge distribution:

$$
\begin{aligned}
P(x_{1..D-1}) &= \mathcal{N}(0, \sigma_1) \\
P(x_D) &= \mathcal{N}(y + 3 * y + (1 - y)^2, \sigma_2) \\
y &= \sum_{i=1}^{D-1} x_i
\end{aligned}
$$

As shown in Fig 3C, also in this case Gaussian processes give the best approximation in two dimensions, but at higher dimensions mixtures of factor analyzers outperform all other methods, showing the value of the dimensionality reduction introduced by the factor analyzers. In two dimensions, kernel densities with full covariance bandwidth matrices are better here than copulas.

**Heavy tails.** A third difficulty in posterior distributions is heavy tails; in this case there will be relatively few samples spread over a large space, making it more difficult to obtain an accurate approximation. To test how well the approximation methods can deal with this, we used a multivariate $t$-distribution with five degrees of freedom as target distribution, with a random covariance matrix as before:

$$
P(\mathbf{x}) = t(\mathbf{x}|\mu = 0, \Sigma, \nu = 5)
$$

Again, Gaussian processes are most accurate in two dimensions. However, in this case a Matérn kernel is better than a squared exponential kernel, as could be expected given the heavier tail of a Matérn kernel (with $\nu = 32$). At higher dimensions, all of the approximation methods have difficulties approximating this distribution.

## Approximating a posterior distribution

To test how the methods perform in approximating a posterior density function, we turned to a dynamic model of a predator-prey system. Specifically, we used a modified Lotka-Volterra system to model the interactions between the Canadian lynx and the showshoe hare [32]. This system was chosen because of the availability of several datasets, a modest number of parameters (5 dynamic parameters and 2 initial conditions for each dataset), and non-linearity in the system which likely leads to non-linearity in the posterior probability distribution of the parameters, making for a meaningful test case.

The model is given by the differential equations

$$\frac{dx}{dt} = \alpha x - (\beta_{\mathrm{kill}} + \beta_{\mathrm{stress}})xy$$
$$\frac{dy}{dt} = \delta xy - \gamma y$$

,

where $x$ represents the hare population and $y$ the lynx population. The populations are measured by their density, i.e. the number of individuals per area in arbitrary units. In the standard Lotka-Volterra model, there is a single parameter $\beta$ for the effect of predation. We have split this effect into two parts, $\beta_{\mathrm{kill}}$ and $\beta_{\mathrm{stress}}$, because it has been shown that at high lynx densities, the hares not only die from increased predation, but also produce less offspring, which appears to be due to stress induced by the constant threat of predation [32, 33]. The modeled natality (number of offspring per adult female in one breeding season) is given by $2 \cdot \exp(\alpha - \beta_{\mathrm{stress}} y)$.

All of the parameters should be positive. To simplify the inference and approximations, we first infer the parameters on log scale, so that there are no discontinuities in the posterior density (we lift this restriction of unbounded priors later). As prior we take a diagonal multivariate normal distribution. When wide priors are used, unrealistic parameter values can be found, corresponding to oscillations through the data points at very high frequency; we therefore restricted the prior to a relatively narrow distribution so that only the correct oscillation with a period of roughly 10 years is obtained.

We used two datasets to infer the parameters. The first dataset is the Hudson Bay Company data of lynx pelt records [34], which we will refer to as the *lynx data*; in particular we used the McKenzie River station data from 1832 through 1851. The second dataset is a study of a hare population and its reproductive output [35], from 1962 through 1976, which we will refer to as the *hare data*. Note that the lynx data only contains measurements of the lynxes while the hare data only contains measurements of the hares. The lynx and hare densities are normalized to be between 0 and 1 by dividing by the maximum observed value. For the likelihood we take normally distributed error with fixed $\sigma$ of 0.15 for the normalized densities and 2.0 for the untransformed natality. The data are obtained from different geographical regions and in different time periods. It is therefore safe to assume that the datasets are independent. The model describes the common predator-prey interaction irrespective of the geographical region and time period. The differences between the datasets are modeled by having separate variables for the initial conditions; i.e. the 5 dynamics parameters are common to the datasets, and for each dataset there are 2 additional parameters for the initial conditions.

We fitted the model to each dataset separately and to the two datasets together; Fig 4 shows the data and the posterior predictive distributions. The model can adequately describe these datasets, both separately and jointly, as evidenced by the good overlap of the posterior predictive distribution and the observed data.

Fig 5A–5C shows several aspects of the posterior obtained after seeing the lynx data. The posterior distribution appears to be unimodal (Fig 5A) and there are correlations between some of the parameters (Fig 5B). The distribution also deviates from a Gaussian distribution as shown by the bivariate scatter plot for two of the parameters (Fig 5C).

We then tested by cross-validation how well the approximations can describe the posterior distribution of the two datasets (see Fig 5D). For the lynx dataset, mixtures of factor analyzers give the best performance; while for the hare dataset Gaussian mixtures and vine copulas with mixture marginals also give good cross-validation performance (Spearman correlation $\rho \approx 0.9$ and the lowest root mean square error).

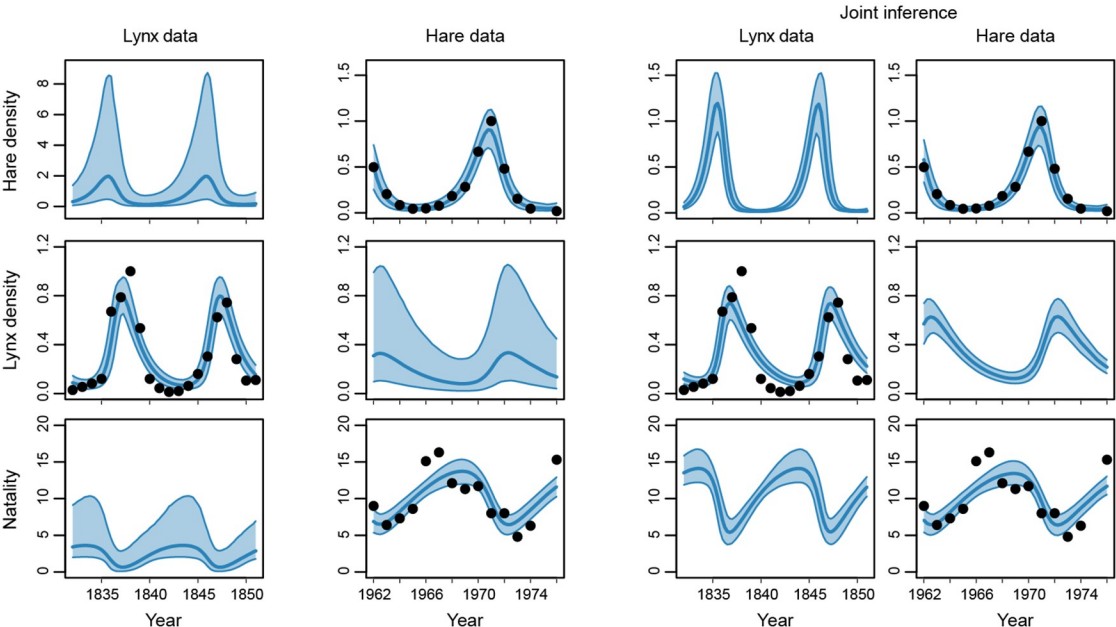

**Fig 4. Lynx-hare datasets and posterior predictive distributions.** The lynx data provides an estimate of lynx density (number of animals per surface area) and the hare data provides an estimate of hare density and natality. Black dots indicate the data, the thick blue line is the median and the shaded blue area the 90% confidence interval of the posterior predictive.

## Sequential inference

Having obtained reasonably accurate approximations of the posterior densities, we can test how they perform in sequential inference. To do this, we approximated the posterior from the lynx dataset with all methods using 1,000 samples, and use these approximations as prior for inference with the hare dataset. If the approximations are accurate, the resulting posterior of the second inference should give the same result as a joint inference with the two datasets together.

Fig 6A shows the marginal probability density of one of the parameters, $\beta_{\text{kill}}$, from the datasets separately, the true joint, and with two approximation methods (importance reweighting and a gaussian mixture). As expected, importance reweighting provides a very poor approximation; a single sample receives almost all of the weight and the true joint posterior cannot be accurately estimated from essentially one sample. The Gaussian mixture approximation on the other hand provides a sequential posterior that is visually almost indistinguishable from the true joint. To quantify the performance, we calculated the Kolmogorov-Smirnov statistic for the marginal distribution of each of the parameters, based on the marginal empirical cumulative distributions (see Fig 6B and 6C). Both Gaussian mixtures and vine copulas give sequential posteriors that are closest to the true joint. Gaussian processes and the KD approximation perform worse, as expected given their poorer cross-validation performance.

## Marginal likelihood estimation

We can use the posterior distribution approximations to obtain an estimate of the marginal likelihood directly from the Monte Carlo samples (see Methods section). Table 1 shows the estimates obtained from three dedicated marginal likelihood estimation algorithms, compared to the estimates obtained directly from the samples using the posterior approximations. The

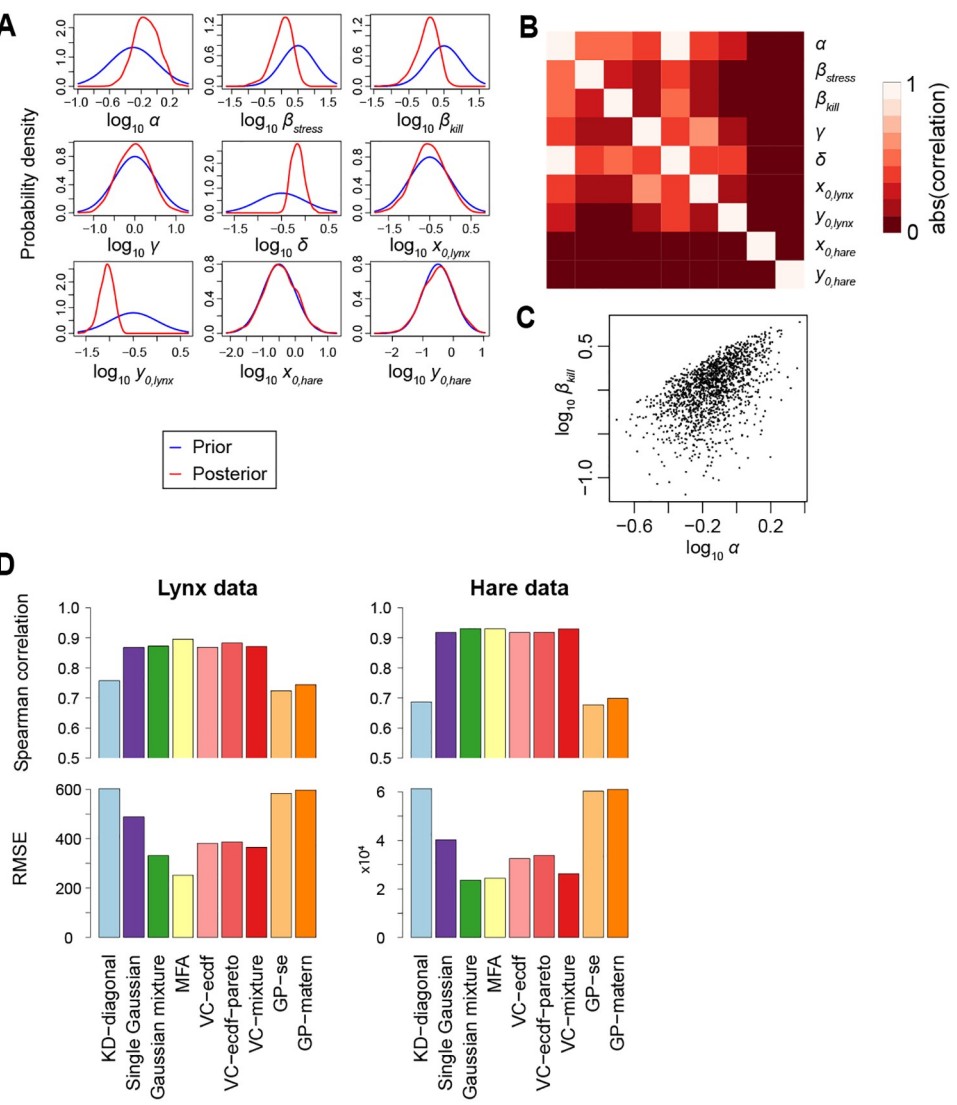

**Fig 5. Lynx and hare dataset posterior approximation.** (A) Marginal posterior densities after seeing the lynx data, the graphs are constructed using kernel density estimation with plug-in bandwidth selection. (B) Correlations between the parameters in the lynx posterior. (C) Scatter plot of the samples for one parameter combination. (D) Approximation accuracy as a function of sample size. Spearman correlation and root mean square error were calculated by comparing the approximation with another 1,000 MCMC samples from the target.

posterior approximations that performed well in cross validation and sequential inference also provide accurate marginal likelihood estimates.

## Bounded priors

In practical applications, it is often the case that the prior probability distribution has a bounded domain, due to known constraints in any of the variables of interest. Some of the approximation methods can handle bounded distributions directly. Alternatively, the variables can be transformed to an unbounded domain (see Methods section). To test these options, we take the same predator-prey model, now inferring the parameters on natural scale and with uniform priors, thus resulting in hard bounds on both the prior and the posterior distribution.

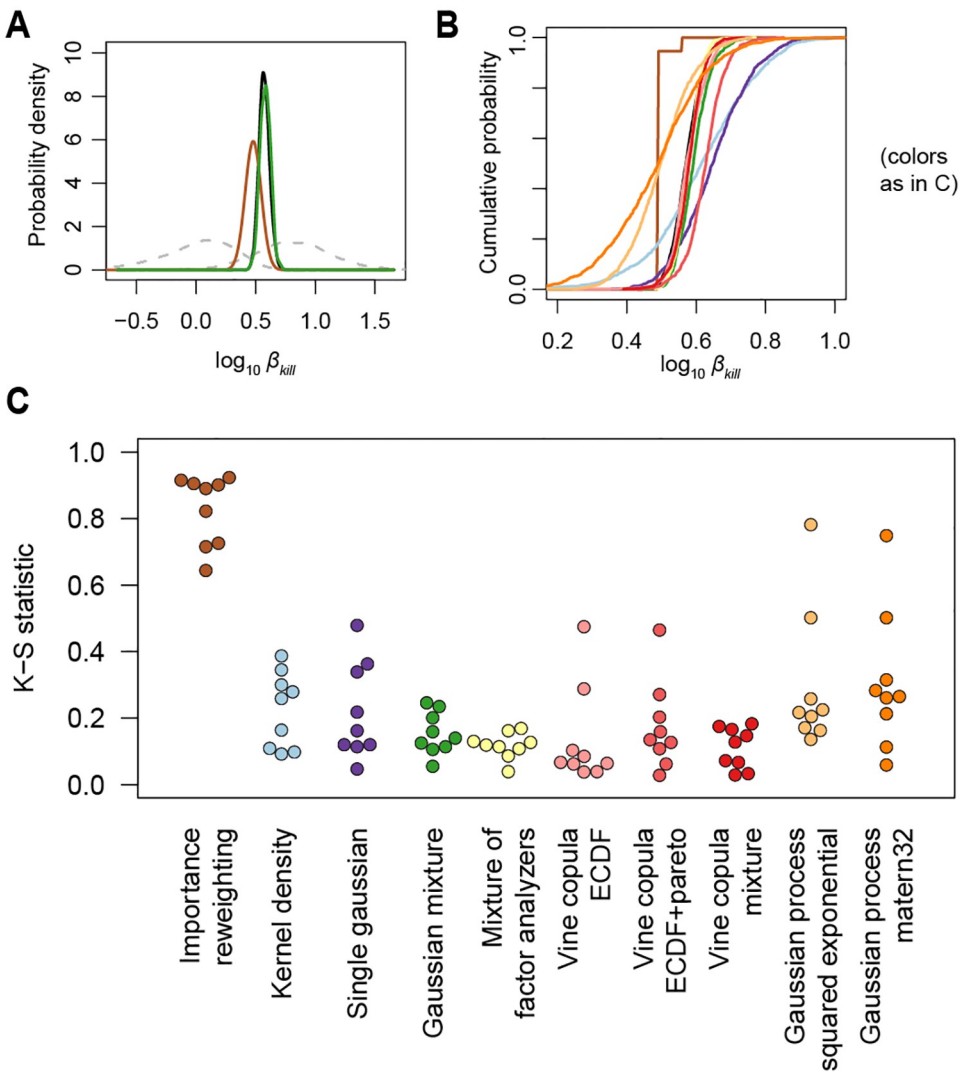

**Fig 6. Sequential inference performance.** (A) Marginal density of one of the parameters (for clarity, only the GM approximation and importance reweighting result is shown). The dashed lines indicate the posterior of the two datasets separately, and the black line is the true joint. Other colors are the same as in C. (B) Empirical cumulative distribution of the same parameter, showing all approximation methods. The colors are the same as in C. (C) Kolmogorov-Smirnov statistics for the comparison of the marginal distributions of the true joint to the marginals of the posterior obtained after sequential inference with each of the approximation methods. Each dot indicates one of the parameters.

**Table 1. Log marginal likelihood estimates (± estimation variance if available).**

| Method | Lynx data | Hare data |
|---|---|---|
| Thermodynamic integration | 0.46±0.92 | −34.7±1.4 |
| Sequential Monte Carlo | 0.57±0.42 | −34.7±0.36 |
| Nested sampling | 0.77±0.65 | −34.4±0.64 |
| Kernel density estimate | 4.60 | −29.1 |
| Gaussian mixture | 0.80 | −34.5 |
| Vine copula—mixture | 1.20 | −34.3 |
| Gaussian process—SE | 5.19 | −28.8 |

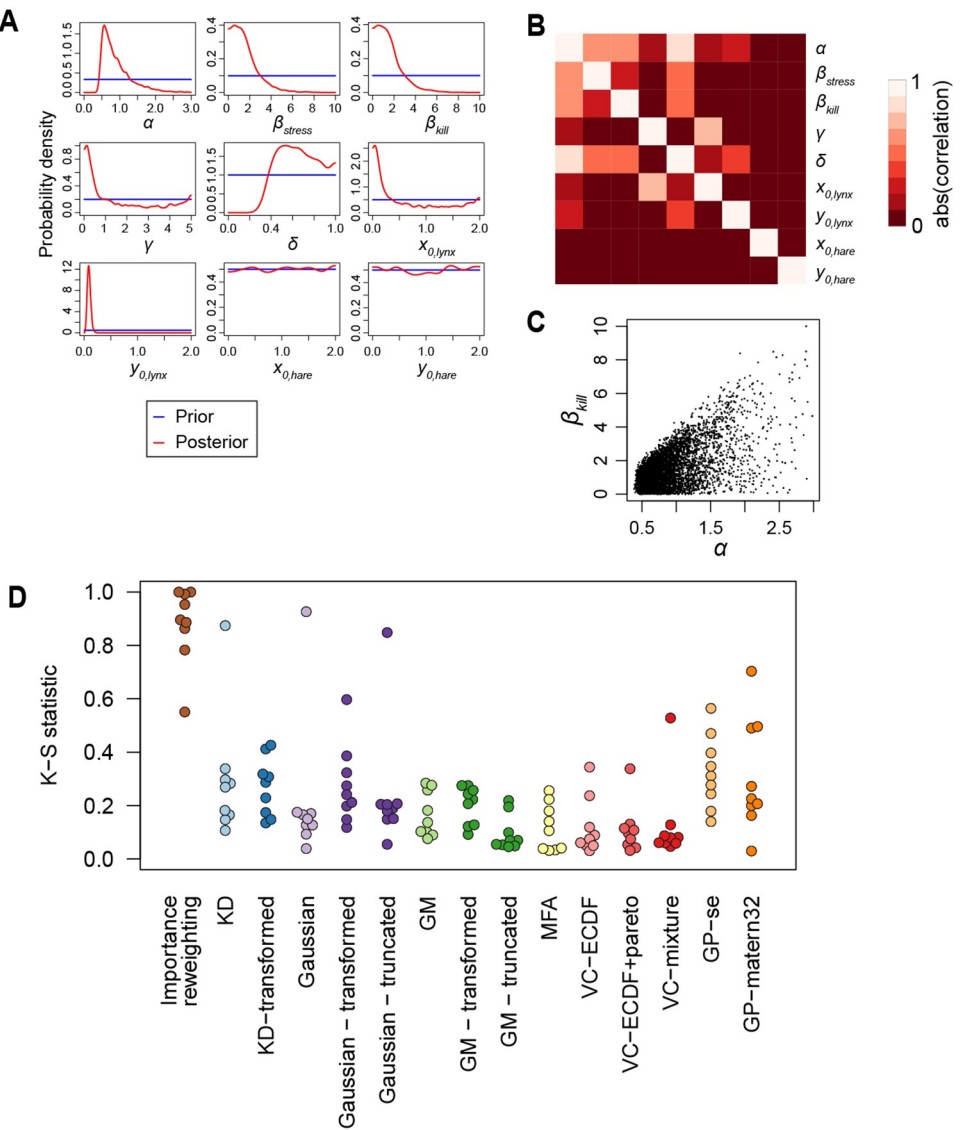

**Fig 7. Sequential inference with bounded priors.** (A) Marginal posterior densities after seeing the lynx data; compare with Fig 5A. (B) Correlations between the parameters in the lynx posterior. (C) Scatter plot of the samples for one parameter combination. (D) Sequential inference accuracy; same as in Fig 6, with the addition of transformed and truncated variations.

As before, the prior is chosen such that only the correct oscillation with a period of 10 years is obtained.

Fig 7A–7C shows several aspects of the posterior distribution of the lynx data, as before in the log-transformed setting. It is clear that the bounds on the prior distribution leads to a large discontinuity in the posterior probability distribution at the bounds for most parameters. The sequential inference test (Fig 7D) shows that for KDs and GMs, it is beneficial to specifically handle these boundaries; either by variable transformation or using truncated Gaussians in the case of Gaussian mixtures. For vine copulas, the marginal transformations can handle bounded domains, but the performance is nevertheless worse than in the unbounded situation before.

### Efficiency of sequential versus joint inference

One of the motivations for using posterior approximations and sequential inference was that it may allow a computationally faster evaluation of the joint posterior. For evaluating the posterior of a first dataset, the likelihood of the second dataset does not need to be evaluated and vice versa. More importantly, some of the parameters may only be relevant for one of the datasets and could thus be dropped from the inference, thereby reducing the dimensionality of the inference.

To test this, we compared the accuracy of the posterior obtained from a joint inference to the posterior obtained by sequential inference, given a fixed total number of model evaluations. These posteriors are in turn compared to a "ground-truth" obtained from the joint inference with 100-fold more model evaluations. As test system we used the unbounded Lotka-Volterra system described above. In this case the two separate inference steps in the sequential inference route contain only 7 parameters (5 kinetic parameters + 2 initial conditions), whereas the joint inference has 9 parameters (5 kinetic parameters + 2 × 2 initial conditions), so the sequential inference should have an advantage in sampling efficiency due to the lower dimensionality. We used Gaussian mixtures as posterior approximations.

Fig 8 shows the mean and standard deviation of the joint posterior distribution of the five kinetic parameters obtained in 10 separate runs. Each run was entirely separate; in each run new sampling was done from the first posterior (including new starting points for the MCMC chains and the hyperparameters of the sampling algorithm were optimized separately). New approximations were then fitted to these samples, and the approximations are used as prior for a new round of sampling with the second dataset. From Fig 8, it is clear that sequential inference by sample reweighting introduces a large bias and variance and is not a viable means of obtaining the joint posterior. Using a Gaussian mixture approximation after the first inference greatly improves the accuracy compared to sample reweighting. Nevertheless, the posterior obtained from sequential inference is less accurate than the posterior obtained from joint inference. For example, for the parameter $\beta_{kill}$, the sequential inference introduces either more variance (when the lynx dataset is used first), or a slight bias (when the hare dataset is used

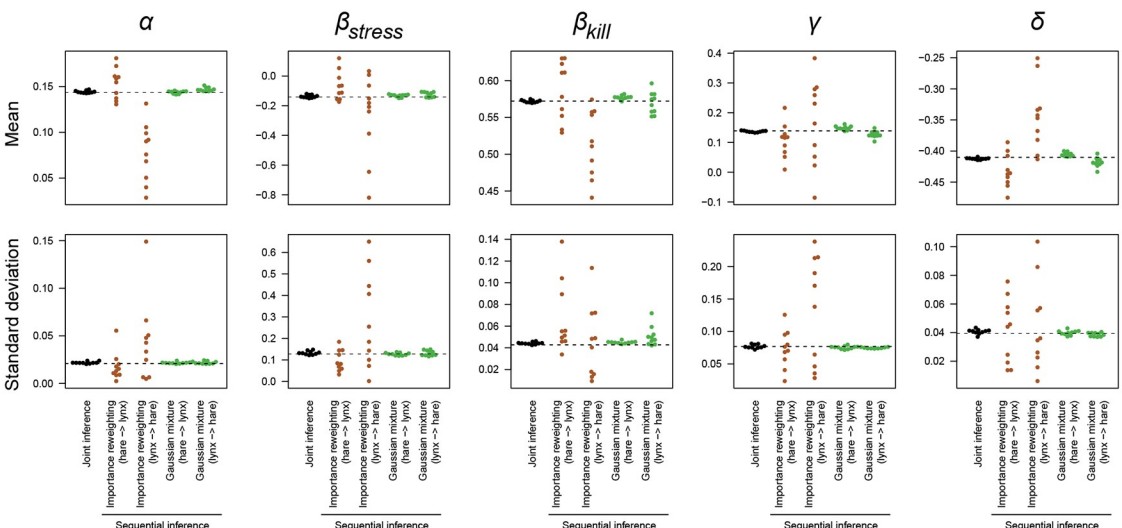

**Fig 8. Accuracy of joint versus sequential inference.** Each point represents the mean and standard deviation of the marginal posterior distribution from one run. Each run has the same total number of model evaluations (1.8 million). The dashed line indicates the mean and standard deviation of the posterior from a joint inference run with 100-fold more model evaluations.

**Table 2. Time complexity of training and evaluation of the approximation methods.** Evaluation is the cost of evaluating one new sample. $N$ = number of Monte Carlo samples used for the estimation, $D$ = dimensionality, $G$ = number of mixture components. The training time gives the number of seconds required to fit a 10-dimensional approximation on 1,000 samples using *mvdens*. For mixture fitting, the time for fitting a 5-component model is reported; optimizing the number of components will grow linearly with the number of components considered.

| Method | Training | Evaluation | Training time (s) |
|---|---|---|---|
| Kernel density estimate | $N^2D$ | $ND$ | <1 |
| Gaussian mixture | $G^2ND^3$ | $GD^2$ | 4 |
| Truncated Gaussian mixture | $G^2ND^3$ | $GD^2$ | 2521 |
| Mixture of factor analyzers | $G^2ND^3$ | $GD^2$ | 21 |
| Vine copula—ecdf | $N^2D + ND^2$ | $ND + D^2$ | 104 |
| Vine copula—mixture | $G^2ND + ND^2$ | $GD + D^2$ | 168 |
| Gaussian process | $N^3 + D$ | $ND$ | 370 |

first). During sequential inference, the increased sampling efficiency in the individual inference steps does not outweigh the error introduced by the intermediate posterior approximation.

## Time complexity

The approximation methods differ in the computational cost of training and evaluation. Table 2 lists the time complexity of each method.

Typically, the number of Monte Carlo samples $N$ will be (much) larger than the dimensionality $D$. Since Gaussian mixtures and vine copulas with mixture marginals do not depend on the number of samples during evaluation, they can achieve the fastest performance when a large number of evaluations are needed in the sequential inference. Kernel density estimates, Gaussian processes and vine copulas with empirical marginals do depend on the number of samples and can thus be significantly slower when a large number of samples is used.

Gaussian processes have cubic scaling with respect to the number of samples for the training, which severely limits the number of samples that can be used. While there are approximation methods available for GPs with large input sizes [11], the use of GPs for posterior approximation appears to be best suited for low $N$ and $D$.

## Failure case

To illustrate the present limits of this approach to sequential inference, we also discuss a case where the approximations fail to provide an accurate posterior.

A more challenging test case is given by a model of biological signaling in cancer cells. The goal here is to explain how different breast cancer cell lines respond to kinase inhibitors by modeling how the signal arising from oncogenic driver mutations is propagated through a signaling network. These models are constructed using feedback-Inference of Signaling Activity and are described in more detail in [5, 36]. Here we will use a small test model, which is shown graphically in Fig 9A and the resulting equations are given below. Briefly, the model contains four observed variables, namely the ERBB2 amplification status, PIK3CA mutation status and phosphorylation of AKT and PRAS40 (represented by **m**, **n**, **p** and **q** respectively). The amplification and mutation status is known with certainty, so the variables are directly set to 1 if the amplification or mutation is present and 0 otherwise. The remaining three variables, PI3K activation, AKT activation and PRAS activation (represented by **x**, **y** and **z** respectively) are latent variables, and the inhibitor concentration $w$ is given.

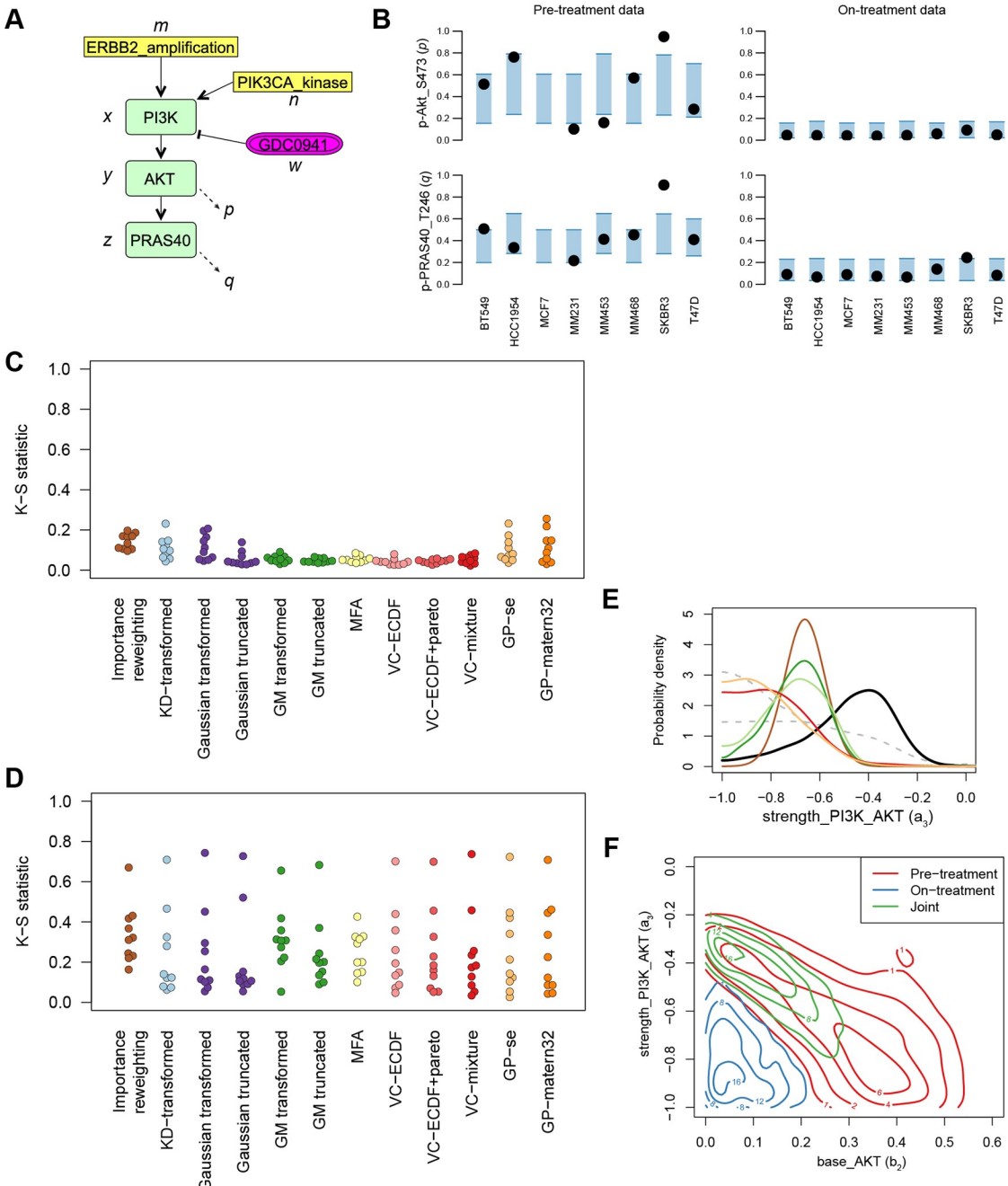

**Fig 9. Sequential inference in the breast cancer signaling model.** (A) Signaling model in Systems Biology Graphical Notation format. (B) Data and posterior predictive distributions. Black dots indicate the data and the blue shaded area is the 90% confidence interval of the predictive mean. "p-Akt_S473" is the measurement of $p$ and "p-PRAS40_T246" is the measurement of $q$. (C) Performance in sequential inference when the data is split by first using the measurement of $p$ and then $q$ (i.e. first use the data shown in the top two graphs in (B), and then the bottom two). (D) Performance in sequential inference when the data is split by pre-treatment and on-treatment (i.e. first use the data shown in the left two graphs shown in (B), and then the right two). (E) Density of one of the parameters as inferred by joint inference (black line) and through sequential approximation split by treatment (colored lines). The grey lines indicate the separate posteriors of the pre-treatment and on-treatment data. (F) Contour plot of the bivariate posterior density of two of the parameters obtained from either dataset alone or the true joint.

The model is described by the equations

$$\mathbf{x} = f(b_1 + a_1\mathbf{m} + a_2\mathbf{n}) \cdot g(w)$$

$$\mathbf{y} = f(b_2 + a_3\mathbf{x})$$

$$\mathbf{z} = f(b_3 + a_4\mathbf{y})$$

$$P(\mathbf{p}|\mathbf{y}) = t(\mathbf{p}|\mu = \mathbf{y}, \sigma = 0.2, v = 3)$$

$$P(\mathbf{q}|\mathbf{z}) = t(\mathbf{q}|\mu = \mathbf{z}, \sigma = 0.2, v = 3),$$

where

$$f(\mathbf{x}) = 1.0/(1.0 + \exp(-9.19024(\mathbf{x} - 0.5)))$$

$$g(w) = k + (1 - k)/(10^{s(w-h)} + 1)$$

and $t$ is Student's $t$-distribution with fixed $v = 3$ and $\sigma = 0.2$. The remaining 10 variables are parameters to be inferred. The strength parameters $\mathbf{a}$ are estimated on a log-10 scale with a uniform prior and the remaining parameters are estimated on a regular scale, also with a uniform prior [36]. The measurements have been normalized to take values between 0 and 1.

To test whether the sequential inference gives a good approximation also in this setting, we study sequential inference by incorporating parts of a dataset sequentially. The dataset contains measurements of protein phosphorylation without drug treatment (referred to as the pre-treatment data), as well as after 30 minutes of drug treatment (referred to as the on-treatment data), in eight cell lines (see Fig 9B). The drug concentration $w$ is 0 in the pre-treatment setting and 1 μM in the on-treatment setting.

We first test sequential inference in the same way as for the lynx-hare model, by splitting the data by observable. That is, we first infer the posterior with observations of $\mathbf{p}$, and subsequently update the posterior with observations of $\mathbf{q}$. As can be seen in Fig 9C, sequential inference performs well in this case. The observations of $\mathbf{q}$ are correlated with $\mathbf{p}$, and so the first posterior is only slightly refined by the further inclusion of $q$ (in most dimensions).

A potentially more useful sequential inference would be to split the dataset in a pre-treatment and on-treatment set. That is, we would first use the observations of both $\mathbf{p}$ and $\mathbf{q}$ for $w = 0$ and then for $w = 1$. The accuracy of the sequential inference when split in this way is shown in Fig 9D. Unfortunately, none of the approximation methods gives posterior distributions that agree with the joint inference. For several parameters the resulting empirical distributions always have a large discrepancy. Fig 9E shows this in more detail for one of the parameters. When investigating this poor performance, we found that this is due to the pre- and on-treatment parts of the data inducing widely different posteriors. As shown in Fig 9F, the pre- and on-treatment data are essentially contradictory for the parameters $b_2$ and $a_3$: the on-treatment data indicates low values for both parameters, whereas the pre-treatment data indicates higher values. The model can still reconcile these data, as the joint inference shows that a high strength $a_3$ is favored when both datasets are included. To recover this joint posterior using approximations would require that the approximations are highly accurate in the tails of the posterior of the pre-treatment data. But standard Monte Carlo methods, and by extensions the approximation methods based on them, are typically not well suited for estimating the tails of a distribution, since most samples will be concentrated in the body of the distribution. Sequential inference with posterior approximations therefore seems to be unsuitable when the separate datasets give rise to strongly divergent posterior distributions.

## Discussion

When using sequential Bayesian inference in combination with Monte Carlo sampling, we are restricted to using samples from a first inference as prior for a second inference. This can be done by directly reweighting the samples, or by approximating a functional form of the posterior distribution from the Monte Carlo samples. We have explored the use of several such approximation methods, and we found that they can allow more accurate sequential inference than direct sample reweighting.

The approximation methods have different strengths and weaknesses. Gaussian processes are highly efficient in low dimensionality, but they deteriorate in higher dimensions, at least when using isotropic kernels. Both Gaussian mixtures and vine copulas can give good approximations also in higher dimensions. Vine copulas do not work well for multimodal distributions however. Kernel density estimation appears to be less efficient than the other methods. Finally, none of the approximation methods we tested are adequate in the far tails, although this is more likely to be a limitation of the Monte Carlo sampling rather than the approximation methods, as by definition the tail only contains a small part of the Monte Carlo samples.

In this work we have focused on models with relatively low dimensionality, with examples and test cases containing up to 10 dimensions. In many cases of applied Bayesian analysis models with significantly more dimensions are considered [2, 3, 5], and in future work it would be important to explore how the approximation methods extend to higher dimensionality. If the trends observed in the examples with known target distributions extend beyond 10 dimensions, we would expect that methods which employ dimensionality reduction, like factor analyzers or methods based on them, would be most useful in higher dimensional settings.

Many further extensions to the posterior approximation methods can be considered. Using mixtures of t-distributions could improve upon Gaussian mixtures in estimating the tails of the distributions [37, 38]. For vine copulas, the approximation of the marginal distributions can have a strong effect on the accuracy. Further improvements for marginals using Pareto tails could be achieved by estimating an optimal Pareto tail threshold instead of using a fixed value, and estimating the body and tail distributions together [39, 40]. Given the good performance of Gaussian process regression in lower dimensions, it would be interesting to explore how this can be better extended to higher dimensions. Using anisotropic kernels will likely be beneficial, but this introduces additional parameters that need to be optimized during the regression. To make this computationally feasible it would be necessary to use approximations to the GP, e.g. [11, 41]. For kernel density estimates, sparse covariance matrices merits exploration as well, for example the method proposed by Liu *et al.* [42]. For copulas, it would also be interesting to explore multimodal extensions, such as the method proposed by Tewari *et al.* [43].

An alternative approach could be to use variational inference rather than Monte Carlo sampling. In the present context of sequential inference it would make sense to estimate a functional form of the posterior directly during inference (i.e., do variational inference), rather than first sampling and then estimating a functional form of the posterior from the samples. There has been recent progress in variational inference with Gaussian mixtures with full covariance matrices [44, 45]. Given that Gaussian mixtures can provide good approximations in our test cases, this would be an interesting avenue to explore further, although the matrix computations involved in these variational inference methods still pose challenges in higher dimensions.

There are various reasons why it might be useful to do sequential inference. Sequential inference can be conceptually appealing: all information relevant for the model is stored in the posterior distribution, allowing the modeler to discard a dataset after the inference.

Additionally, sequential inference allows us to update the posterior of an existing model when new data or samples become available, even when the initial data is no longer available. This can also be useful when an inference task was computationally demanding, in which case it may be impractical to redo a joint inference when additional data becomes available.

Nevertheless, sequential inference using intermediate posterior approximations from Monte Carlo samples is an approximation to the joint inference which can introduce bias or additional variance in the joint posterior estimates. In our Lotka-Volterra test case the posterior obtained from sequential inference was accurate, but a joint inference was still more efficient. In the test case of signaling in cancer cells, sequential inference introduced a large bias and hence resulted in wrong joint posterior estimates. Whenever Monte Carlo sampling is used for inference with multiple datasets, joint inference appears to be preferable over sequential inference.

## Supporting information

**S1 File.**
(TSV)

## Author Contributions

**Conceptualization:** Bram Thijssen, Lodewyk F. A. Wessels.

**Data curation:** Bram Thijssen.

**Formal analysis:** Bram Thijssen.

**Funding acquisition:** Lodewyk F. A. Wessels.

**Investigation:** Bram Thijssen.

**Methodology:** Bram Thijssen.

**Resources:** Lodewyk F. A. Wessels.

**Software:** Bram Thijssen.

**Supervision:** Lodewyk F. A. Wessels.

**Validation:** Bram Thijssen.

**Visualization:** Bram Thijssen.

**Writing – original draft:** Bram Thijssen.

**Writing – review & editing:** Bram Thijssen, Lodewyk F. A. Wessels.

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
