## [Decision Letter · Decision Letter 0]

18 Aug 2019

PONE-D-19-17579

Approximating multivariate posterior distribution functions from Monte Carlo samples for sequential Bayesian inference

PLOS ONE

Dear Thijssen,

Thank you for submitting your manuscript to PLOS ONE. After careful consideration, we feel that it has merit but does not fully meet PLOS ONE’s publication criteria as it currently stands. Therefore, we invite you to submit a revised version of the manuscript that addresses all points raised by both reviewers.

We would appreciate receiving your revised manuscript by Oct 02 2019 11:59PM. To enhance the reproducibility of your results, we recommend that if applicable you deposit your laboratory protocols in protocols.io, where a protocol can be assigned its own identifier (DOI) such that it can be cited independently in the future. For instructions see: http://journals.plos.org/plosone/s/submission-guidelines#loc-laboratory-protocols

We look forward to receiving your revised manuscript.

Kind regards,

Hans A Kestler

Academic Editor

PLOS ONE

Journal Requirements:

Reviewers' comments:

Reviewer's Responses to Questions

**Comments to the Author**

1. Is the manuscript technically sound, and do the data support the conclusions?

Reviewer #1: Yes

Reviewer #2: Yes

2. Has the statistical analysis been performed appropriately and rigorously? 

Reviewer #1: Yes

Reviewer #2: Yes

3. Have the authors made all data underlying the findings in their manuscript fully available?

Reviewer #1: Yes

Reviewer #2: Yes

4. Is the manuscript presented in an intelligible fashion and written in standard English?

Reviewer #1: Yes

Reviewer #2: Yes

5. Review Comments to the Author

Reviewer #1: I enjoyed reading this manuscript, which is written very clearly and focuses on a simple idea using existing tools. The emphasis is primarily on splitting a dataset and then running posterior sampling on the first part of the dataset, obtaining an analytic approximation of the posterior and using this as the prior in the analysis of the second part of the dataset. I do have some major concerns and comments:

(1) The problem of scalable computation for large datasets is quite different from the problem of analyzing multiple datasets in that it is important to account for differences across the datasets in the multiple dataset case. There may be heterogeneity in the parameters which crucial needs to be taken into account and there is a rich literature on associated hierarchical models and methods for meta analysis. This type of issue is not discussed at all in the paper, and I would worry that the naive reviewer may use the proposed method not realizing that it is effectively the same as pooling the different datasets together and using a single statistical model which assumed observations from different datasets are exchangeable. This needs to be properly addressed.

(2) If the focus is instead entirely on accelerating computation for a single large dataset by splitting the data into smaller datasets that are analyzed separately in a sequential manner, then there is a large relevant literature that needs to be properly discussed. Currently the authors just cite a single article based on using a product equation with kernel approximations plugged in - this method they cite actually has quite poor performance relative to others as discussed in more recent publications in this area. One needs to motivate why analyze the data sequentially instead of in an embarrassingly parallel manner, which would seem to be faster. Hence, the practical motivation & support relative to the literature for the method is quite lacking.

(3) A very very major issue that is not properly discussed in my view is the curse of dimensionality and the fact that the proposed methods are not going to do well at all when the number of parameters in the model is not quite small - say about 5 with 10 being too large most likely. Most interesting Bayesian models have many-many more parameters than this & this limits the scope of the methods. The problem is that the posterior approximation methods break down rapidly in dimensionality and the authors really don’t consider methods that are particularly scalable in terms of density estimation - e.g., mixtures of factor analyzers and such; see “Geometric density estimation” work by Wang, Canale and Dunson for example. One seemingly big motivation for not taking a sequential approach is that one can run posterior computation in parallel for chunks of the data & then combine in a targeted manner for low-dimensional functionals and parameters of interest instead of combining for the entire high-dimensional posterior; see, for example, Li, Srivastava and Dunson 2017 Biometrika.

(4) If you do have a setting with very small numbers of parameters in the model and very large sample size (motivating the proposed approach) then it would seem that simpler methods are well justified - for example, Laplace approximation and Gaussian approximations to the posterior would be accurate.

Hence, I’m wondering if there is any real motivation for the proposed approach relative to the literature.

Reviewer #2: In the manuscript "Approximating multivariate posterior distribution functions from Monte Carlo samples for sequential Bayesian inference", Bram Thijssen and co-workers present an evaluation of sequential inference in the context of Bayesian inference. This manuscript evaluates different methods for the approximation of posterior distributions such as kernel density estimation, Gaussian mixtures and vine copulas. The authors test the results of the approaches on two examples with real datasets, and compared to classical joint inference.

The topic of Bayesian inference is very relevant and there are many open questions. Bayesian inference is often achieved with Monte Carlo sampling approaches. One of the key problems is that Monte Carlo sampling can be very computationally demanding for complex models, therefore methods to reduce this bottleneck are needed. In this manuscript, the authors evaluate this problem from the experimental data perspective by assessing different methods for sequential estimation of the posterior distribution from the Monte Carlo samples of consecutive and independent datasets.

The key contributions are in my opinion (1) collecting and testing many methods for estimating a functional approximation of the posterior distribution and (2) the development of a reusable open-source toolbox/library. This evaluation is in my opinion a valuable contribution. I appreciate that the authors provide reusable code, but I have to admit that I did not have the time to test it. To my mind, it is very valuable that the authors also discuss and demonstrate the limitations of the approaches.

======

MAJOR

======

- In the introduction the authors refer to the case of challenging Bayesian inference for complex models (line 8). What do the authors consider here as a "complex model"?

- It would be great if the authors could comment on the use of the methods for high-dimensional inference, e.g. Hug et al., High-dimensional Bayesian parameter estimation: case study for a model of JAK2/STAT5 signaling. Mathematical Biosciences, 246(2), 293-304, 2013.

- The authors only present one model in which the results are successful, are there other examples that could be included? In particular additional artificial examples would in my opinion be helpful.

- The authors mention that "Estimating functional forms of posterior distributions ... is an established part of ... in the context of estimating orbital eccentricities in astronomy". Are there any other fields in which this is currently used? e.g. Eriksson et al., Uncertainty quantification, propagation and characterization by Bayesian analysis combined with global sensitivity analysis applied to dynamical intracellular pathway models, Bioinformatics, 35(2), 2019.

- Between line 16 and 17: "In our modeling of biological systems" The authors mention the computational challenge they encountered regarding inference with multiple datasets. I think this is a very common challenge in the community, therefore it would be beneficial to cite additional examples (ideally from other groups) showing this.

- I was missing the discussion of radial basis faction approximations for posterior approximations. This approach has already been applied for posterior approximation(e.g. Fröhlich et al., Radial Basis Function Approximations of Bayesian Parameter Posterior Densities for Uncertainty Analysis. In: Mendes P., Dada J.O., Smallbone K. (eds) Computational Methods in Systems Biology. CMSB 2014. Lecture Notes in Computer Science, vol 8859. Springer, Cham, 2014).

- The performance of several of the approximation methods depend on many implementation details. For example, the authors list the R package VineCopula for the evaluation of VCs. What about the other methods? It would be valuable to clarify this in the text.

- Line 312: As the authors know, it is beneficial to use log parameters instead of natural scale. Can the authors apply the bounded setting to log parameters with broader bounds instead of the natural scale?

- Can the authors provide a computation time comparison for the predator-prey model? Since this model works in the sequential setting, a comparison respect to the joint setting regarding computation time would be great.

- Line 191: In MCMC chain, individual samples are dependent. May I ask how this dependence structure (which is disregarded in the first example) will influence the performance of methods? Is it possible to account for it? How is this related to the required sample size?

- Line 204: As the authors mention, it is not surprising that GM performs best.

I would find it important to test the method on other common topologies, including structures which arise of parameters are non identifiable (see e.g. Raue et al., Joining forces of Bayesian and frequentist methodology: A study for inference in the presence of non-identifiability. Philos T Roy Soc A, 2013).

======

MINOR

======

- Line 32: Maybe the authors can shortly mention why is challenging to meet the constraint (i.e. must integrate to one) as scaling should be always applicable.

- Line 79: Why for D > 4 only the authors only estimate a diagonal matrix? Has this been shown to perform better? What about bootstrapping methods for bandwidth selection instead of plug-in bandwidth?

- Line 214: I am not an expert in the field of copulas, but the authors mention that "the available copula functions are designed to describe the shape of a single model". Do exist any multi-modal copulas (even though not available in the R library)?

- Can the authors clarify why was BIC used for GM while AIC for VC?

- Line 240: The split of parameter beta appears to introduce a structural non-identifiability. I would suggest that the authors mention and discuss this.

- Line 259: How realistic is the assumption of the noise levels? Has this been quantified? If not, I would suggest to estimate sigma along with the dynamic parameters.

- Line 271: Maybe the authors can integrate Fig. 5A-C better into the text.

- Line 341: Could the authors clarify what are the differences between the different realizations? Different starting points, random seed ... ?

- Line 348: For me is not clear to spot the slight bias in \\beta_{kill} when the hare dataset is used first that the authors mention (Fig. 8). Additionally, there is a shift in the parameters \\gamma and \\delta between the two GM scenarios in the sequential setting, any idea why? I would recommend to plot median values instead.

- Line 372: typo "feecback" → "feedback"

- Second model: All the cell lines have the same parameter values? Was the data normalized? Did the authors use log or natural parameters? As Fig. 9E shows negative values for parameter I would assume is log, but is not clear in the text.

- Figure 1: Axis labels are missing for some of the subplots. I recommend using subsections (e.g., A-B-C) and refer to what is seen directly in the caption. The caption is rather generic and I have difficulties to follow the content of the image. Moreover, a legend will help as, e.g., red solid line and/or black dashed line, are not defined. I do not know what is the difference between the middle column and the right column subplots.

- Figure 2: Axis labels are missing, something like \\theta_1 and \\theta_2 (instead of the model parameters for simplicity). In the manuscript kernel density is shortened as "KD" while in the figure is shown as "KDE". Same for Gaussian mixtures ("GM" vs "GMM"). The authors comment that "slight differences ... can be observed as well" maybe can be highlighted? It is difficult to spot.

- Legend Fig. 4: "posterior predictive" → "posterior prediction"

- Legend Fig. 4: It is unclear for me why the authors normalize again with the maximum. The data are already relative. If the normalization is used, I think that it would be important to clarify how the simulations are normalized. Also with the measured maximum, or with the simulated maximum?

- Fig. 5: As the choice of the prior is often random, it would be interesting to see the effect of the prior, in particular what happens if the prior is less informative.

- Fig. 5 (D): This figure is not completely clear for me. How was the comparison performed? Furthermore, I was wondering why the correlations always increase with sample size but the RMSE does not necessarily decrease?

- Caption for Fig. 5: There is a extra "3." in the end.

- Fig 7 (A) comparison of the results with the log10 shown in fig. 5 is difficult, I would suggest to plot in log10 the X axis for a direct comparison and clarify in the caption that the sampling was performed in linear scale.

- Fig. 8: Are the mean and standard deviation values calculated after removing burn-in phase?

- Caption for Fig. 9 (E) definition for the gray dash lines are missing, I assume that it corresponds to the results from the individual fittings (untreated / treated).

6. PLOS authors have the option to publish the peer review history of their article (what does this mean?). If published, this will include your full peer review and any attached files.

Reviewer #1: No

Reviewer #2: No

---

## [Author Response · Author response to Decision Letter 0]

27 Nov 2019

We have made major revisions to the manuscript to address the concerns of the reviewers. Please find a point-by-point reply in the reply to reviewers document which describes all the changes that have been made. We also made sure the figures now comply with the PLoS guidelines.

---

## [Decision Letter · Decision Letter 1]

24 Feb 2020

Approximating multivariate posterior distribution functions from Monte Carlo samples for sequential Bayesian inference

PONE-D-19-17579R1

Dear Dr. Thijssen,

We are pleased to inform you that your manuscript has been judged scientifically suitable for publication and will be formally accepted for publication once it complies with all outstanding technical requirements.

With kind regards,

Alan D Hutson

Academic Editor

PLOS ONE

Additional Editor Comments (optional):

Please attend to the typographical errors noted.

Reviewers' comments:

Reviewer's Responses to Questions

**Comments to the Author**

1. If the authors have adequately addressed your comments raised in a previous round of review and you feel that this manuscript is now acceptable for publication, you may indicate that here to bypass the “Comments to the Author” section, enter your conflict of interest statement in the “Confidential to Editor” section, and submit your "Accept" recommendation.

Reviewer #3: All comments have been addressed

2. Is the manuscript technically sound, and do the data support the conclusions?

Reviewer #3: Yes

3. Has the statistical analysis been performed appropriately and rigorously? 

Reviewer #3: Yes

4. Have the authors made all data underlying the findings in their manuscript fully available?

Reviewer #3: Yes

5. Is the manuscript presented in an intelligible fashion and written in standard English?

Reviewer #3: Yes

6. Review Comments to the Author

Reviewer #3: In the current manuscript, the authors investigate a specific sequential inference approach consisting of approximating the posterior of one data set and subsequently using the approximation as prior for inference on a second data set, based on Monte Carlo sampling. The whole manuscript is pretty well written and organized. Moreover, the issues mentioned by the reviewers in the last round of review have been pretty well addressed. The authors have been able to provide more information where needed and also more explanations concerning some specific points of concern. There are however some minor typos that still have to be corrected:

- page 5 line 107: "... IN MULTIPLE WAYS, ONE OF WHICH IS BY ...": please reformulate (e.g. ... one of which consists of ...)

- page 5 line 108: "D x m": please replace x with \\times (multiplication)

- page 6, 7 lines 128, 134, 147. Please use punctuation marks accordingly (e.g. "Empirical distribution marginal: An empirical distribution function ...")

- page 7, line 161: "... this would ensures ...": please correct accordingly

- page 8, line 175: "... in 5-fold cross-validation ...": please correct accordingly

- Fig 6 A & B: Legend is missing

- page 15, line 382: "... 2*2 ...": Please replace * by \\times (multiplication)

7. PLOS authors have the option to publish the peer review history of their article (what does this mean?). If published, this will include your full peer review and any attached files.

Reviewer #3: No

---

## [Editor Report · Acceptance letter]

2 Mar 2020

PONE-D-19-17579R1 

Approximating multivariate posterior distribution functions from Monte Carlo samples for sequential Bayesian inference 

Dear Dr. Thijssen:

I am pleased to inform you that your manuscript has been deemed suitable for publication in PLOS ONE. Congratulations! Your manuscript is now with our production department. 

With kind regards,

on behalf of

Dr. Alan D Hutson 

Academic Editor

PLOS ONE